# Association of Exposure to Phthalate Metabolites with Antenatal Depression in US Pregnant Women

**DOI:** 10.3390/toxics13100838

**Published:** 2025-09-30

**Authors:** Pallavi Dubey, Chinthana Thangavel, Abdelrahman Yousif, Sophie Kim, Sireesha Reddy

**Affiliations:** Department of Obstetrics and Gynecology, Paul L. Foster School of Medicine, Texas Tech University Health Sciences Center El Paso, El Paso, TX 79905, USA; cthangav@ttuhsc.edu (C.T.); yousif.abdelrahman@ttuhsc.edu (A.Y.); kim72939@ttuhsc.edu (S.K.)

**Keywords:** phthalate metabolites, antenatal depression, pregnant women, patient health questionnaire, composite phthalates

## Abstract

Antenatal depression affects 10–20% of pregnant women, with notable adverse outcomes for the neonates. Limited studies have indicated a potential link between exposure to phthalate metabolites and depression. The association between phthalate metabolites and depression in pregnant women is unknown. We sought to evaluate the association of exposure to phthalate metabolites with depression severity score in US pregnant women. This cross-sectional study used data collected by the National Health and Nutrition Examination Survey during 2005–2018 on pregnant adults who completed urinary profiles that examined 12 common phthalate metabolites. Linear and quantile sum regressions were used to evaluate the association between depressive symptoms (measured by the Patient Health Questionnaire, PHQ-9) and concentrations of phthalate metabolites. A total of 208 women were included in the analysis. These women’s mean (SD) age was 27.42 (5.78) years. We found that all the phthalates were associated with PHQ-9 scores except for mono (carboxyoctyl) and mono-isononyl phthalate. Similar results were observed with the association of high levels of phthalates with mild, moderate, and severe depression (PHQ-9 >4 vs. ≤4). All the phthalate metabolites remained significantly associated with depression scores in the adjusted analysis. Among all considered phthalate metabolites, a combination of MCNP, MBP, MiBP, MnBP, and MEHP contributed to the strongest association with higher depression scores. The relative importance was similar for MCNP (weight = 0.32) and MBP (weight = 0.31), followed by MiBP (weight = 0.12), MnBP (weight = 0.08), MEHP (0.07), and MEP (weight = 0.04) for depression scores. Our findings suggest that pregnant women with high exposure to phthalates are more likely to have higher depressive symptom scores.

## 1. Introduction

Depression poses a significant cause of disability in the United States, impacting up to 10% of adults McCarron, Shapiro [1]. A combination of genetic and environmental factors influences this multifactorial disease. These factors include internal elements like hormones and diet, and external factors such as exposure to environmental contaminants and societal influences [2]. Notably, the National Institute of Mental Health reports a higher prevalence of depression among females compared to males. Females, particularly during hormonal transitions like pregnancy, exhibit increased vulnerability to stress-related and mood disorders [2]. Moreover, postpartum depression affects 20% of women of childbearing age [3]. Approximately one in seven pregnant women meets the criteria for major depressive disorder [4]. Detecting and monitoring depression during pregnancy becomes crucial due to the potential complications it poses for both the mother and the fetus. From a health economics standpoint, the correlation between depression during pregnancy and the perinatal period is linked to heightened utilization of healthcare services, leading to increased expenditures [5]. Numerous studies have established a link between depression during pregnancy and adverse neonatal outcomes, including preterm birth, low weight, developmental delay, behavioral issues, and psychiatric disorders [6]. Environmental stressors play a central role in the development of depression during pregnancy. Considering the intricate pathophysiological nature of depression, investigating modifiable environmental factors could yield valuable insights.

Emerging evidence suggests a potential link between exposure to endocrine-disruptive chemicals (EDCs) and the development of neurological disorders, including depression [7]. EDCs encompass a group of synthetic compounds known to interfere with endocrine function, impacting reproductive health. The broad category of chemicals includes industrial compounds, plasticizers like phthalates and bisphenol A, cosmetics, flame retardants, and pesticides [8,9]. Among them, phthalates are the most abundant EDCs in the environment and daily consumer products. They can be categorized into high molecular weight phthalates (HMWP), found in building materials, medical devices, and food containers, as well as low molecular weight phthalates (LMWP), commonly used in personal care and cosmetic products [10]. Phthalates typically have a short half-life of less than 24 h, making urine analysis an effective method for characterizing their metabolism and excretion [11]. Although phthalate metabolites are present in detectable levels in most individuals, there is a lack of comprehensive research on the connection between exposure to these metabolites and antenatal depression among pregnant women. The objective of this study was to investigate the association between phthalate metabolites in urine samples of pregnant individuals and depression levels. We hypothesized that some HMWP metabolites are associated with depression scores to a greater extent than LMWP metabolites in pregnant women.

## 2. Materials and Methods

### 2.1. Study Population

We conducted a cross-sectional analysis of data derived from the National Health and Nutrition Examination Survey (NHANES), designed to assess individuals’ health and nutritional status in the United States. The NHANES employs a multistage stratified and cluster survey design to represent noninstitutionalized individuals in the country and is conducted by the National Center for Health Statistics and the Centers for Disease Control and Prevention. For this study, we utilized data collected from 2005 to 2018 from NHANES. Out of 17,190 women, we identified 604 pregnant women during survey data collection in NHANES. Following the exclusion of participants without available data on phthalate levels or depression scores, our study included a sample of 208 women. This was primarily due to the collection of phthalate data in one-third of the surveyed subjects in NHANES. The NHANES surveys and examinations obtained written informed consent from all participants after receiving approval from the National Center for Health Statistics Research Ethics Review Board. We followed the Strengthening the Reporting of Observational Studies in Epidemiology (STROBE) reporting guidelines for reporting and analyses.

### 2.2. Exposure Assessment

Thirteen metabolites were quantified using a high-performance liquid chromatography–electrospray ionization–tandem mass spectrometry method. These metabolites include monocarboxynonyl phthalate (MCNP), monocarboxyoctyl phthalate (MCOP), mono-isononyl phthalate (MNP), mono(2-ethyl-5-carboxypentyl) phthalate (MECPP), mono(2-ethyl-5-hydroxyhexyl) phthalate (MEHHP), mono(2-ethylhexyl) phthalate (MEHP), and mono(2-ethyl-5-oxohexyl) phthalate (MEOHP), mono-n-butyl phthalate (MBP), mono (3-carboxypropyl) phthalate (MCPP) of di-n-octyl phthalate (DiNOP), mono-ethyl phthalate (MEP) of di-ethyl phthalate (DEP), mono-isobutyl phthalate (MiBP) of di-isobutyl phthalate (DiBP), monobenzyl phthalate (MBzP). MHNC was not included in the final analysis as more than 70% of the data was missing. We also categorized these phthalate compound concentrations into high and low exposure using the median value of respective compounds for validation analysis.

### 2.3. Depression Score Assessment

The PHQ-9, a 9-item depression screening instrument, was administered to measure the severity of depression symptoms in patients who experienced symptoms over the past two weeks. The PHQ-9 incorporates the Diagnostic and Statistical Manual of Mental Disorders (DSM)-IV depression diagnostic criteria. Response categories for the 9-item instrument are given a point value ranging from 0 to 3 based on the following responses: not at all, several days, more than half the days, and nearly every day. The sum of 9 items is considered as PHQ-9 scores ranging from 0 to 27, which was used for the primary analysis. Further, PHQ-9 scores were categorized as depression (score of >4) or no/minimal depression (score of 0–4), as used in previous studies, to describe the association between depression and other variables.

### 2.4. Covariates

We extracted all potential confounders, including sociodemographic characteristics: age (years), race and ethnicity (Hispanic, non-Hispanic or other [non-Hispanic Asian or other races, including multiracial]), annual household income (<$45,000, $45,000–$99,999 or ≥$100,000), country of birth (US-born or non-US-born), marital status (married or other [widowed, divorced, separated, never married or living with partner]), educational level (less than 9th grade, 9th to 12th grade or college and above)), and health behavior characteristics, including smoking status (yes or no), drinking status (yes or no), physical activity (yes or no), and obesity (yes or no).

### 2.5. Statistical Analysis

Statistical analysis was performed between 21 May 2023 and 19 July 2023. All the data were summarized using appropriate descriptive statistics, including mean with standard deviation (SD), median with interquartile range (IQR), and frequencies with percentages. The association between the PHQ-9 scores and concentrations of each phthalate metabolite was evaluated using unadjusted and adjusted linear regression analyses. The adjusted analyses included all the potential covariates, including age, ethnicity/race, education, marital status, income status, smoking, alcohol use, physical activity, and obesity. We used log-transformed values of phthalate metabolites and depression scores in the regression analysis. The results were summarized using the regression coefficient (RC) with a corresponding 95% confidence interval (CI) and *p*-value. Unadjusted and adjusted weighted quantile sum regression (WQSR) were also conducted to determine the association between mixtures of phthalate metabolites and depression scores. To validate the findings obtained from adjusted WQSR, we performed WQSR and quantile g-computation regression models using the top phthalate metabolites associated with depression scores obtained in WQRS analysis. Weights were generated for the quantile sum of log-transformed phthalate metabolites, guiding the identification of the key phthalate metabolites linked to depression scores. We further validated the association of each of the categorized phthalate metabolites (above median vs. below median concentrations) with depression levels (PHQ-9 scores >4 vs. ≤4) using logistic regression analysis. The results for logistic regression were presented using odds ratio (OR), corresponding 95% CI, and *p*-value. All statistical analyses were conducted using Stata, version 17 (StataCorp LLC, College Station, TX, USA). A 2-tailed *p* < 0.05 was considered statistically significant. Considering the limited data from each cycle, the weights were not included in the survey data analyses.

## 3. Results

A total of 208 currently pregnant female participants were analyzed in this study. The mean (SD) age of the females was 27.42 (5.8) years, with a majority in the age group of 18–30 years (69.7%). Around 23.1% of women had a household income in the first quartile ($0–$44,999), 48.1% were at or above a college level of education, and 17.8% were US-born. About a quarter of these pregnant women (25.5%) were smokers, with 65.9% alcohol drinkers. Obesity was 35.3%, and 15.4% of women reported some kind of physical activity (Table 1). Amongst the pregnant women, the median (IQR) PHQ-9 score was observed as 3 (1, 5), and 31.3% (n = 65) had depression (PHQ-9 score > 4) with a median (IQR) PHQ-9 score of 8 (6, 12). Depression score was observed to be higher in smokers compared to non-smokers (median (IQR): 4 (1, 7) vs. 3 (1, 5)). However, depression scores were not significantly associated with any of the demographic characteristics (Table 1). Of 12 metabolites, 7 had median concentrations above 5 ng/mL. The lowest concentration was noticed for MiNP (median = 0.87; IQR: 0.67, 0.87 ng/mL), while the highest concentration (median = 60.0; IQR: 21.4, 228.7 ng/mL) was for MEP. Phthalate metabolite concentrations were observed to be higher in depressed women compared to no/minimal depressed women (Appendix A).

### 3.1. Unadjusted and Adjusted Associations of Phthalate Metabolites with Depression Scores

In the unadjusted analysis, almost all the phthalate metabolites were significantly associated with depression scores except for MCOP and MNP (Table 2). All the phthalate metabolites were significantly associated with depression scores in adjusted analysis after adjusting for age, ethnicity/race, education, marital status, income status, smoking, alcohol use, physical activity, and obesity, except for mono (carboxyoctyl) and mono-isononyl phthalate. The depression scores were strongly associated with mono-n-butyl phthalate (RC: 0.17; 95% CI: 0.07, 0.26), mono-(3-carboxypropyl) phthalate (RC: 0.15; 95% CI: 0.05, 0.25), and mono-benzyl phthalate (RC: 0.13; 95% CI: 0.04, 0.21) even after accounting for multiple comparisons in the adjusted analyses (Table 3). However, a high level of mono-n-butyl phthalate (RC:0.4; 95% CI: 0.17, 0.63), mono-benzyl phthalate (RC: 0.39; 95% CI: 0.16, 0.63), mono-(2-ethyl-5-oxohexyl) phthalate (RC: 0.34; 95% CI: 0.12, 0.57), mono-isobutyl phthalate (RC: 0.33; 95% CI: 0.08, 0.58) and mono-(2-ethyl)-hexyl phthalate (RC: 0.32; 95% CI: 0.08, 0.55) was also strongly associated with depression scores after adjusting for multiple comparisons (Appendix A). Moreover, the odds of depression were more than 2-fold higher in pregnant women with high exposure to phthalate metabolites compared to low exposure to phthalate metabolites (Figure 1, Appendix A). In addition, high exposure to mono-n-butyl phthalate (OR: 3.53; 95% CI: 1.69, 7.35), mono-(2-ethyl)-hexyl phthalate (OR: 3.34; 95%CI: 1.59, 7.00), and mono-(2-ethyl-5-oxohexyl) phthalate (OR: 3.12; 95% CI: 1.48, 6.55) metabolites was associated with more than 3-fold with high depression levels in adjusted analyses (Figure 1).

### 3.2. Unadjusted and Adjusted Association of Mixtures of Phthalate Metabolites with Depression Scores

Among all considered phthalate metabolites, a combination of MCNP, MBP, MiBP, MnBP, and MEHP contributed to the strongest association with higher depression scores. The relative importance was similar for MCNP (weight = 0.32) and MBP (weight = 0.31) followed by MiBP (weight = 0.12), MnBP (weight = 0.08), MEHP (0.07), and MEP (weight = 0.04) for depression scores (Appendix A). The composite scores of all 12 phthalate metabolites were associated with depression scores in unadjusted (RC = 0.30; 95%CI: 0.13, 0.47, *p* < 0.001) and adjusted (RC = 0.22; 95%CI: 0.03, 0.41, *p* = 0.029) analyses. These associations were unchanged in validation analyses. In fact, a mixture of 5 phthalate compounds (MCNP, MBP, MiBP, MnBP, and MEHP) was highly associated with depression scores (RC = 0.25; 95%CI: 0.12, 0.38, *p* < 0.001) in adjusted analysis using the quantile g-computation method (Table 4).

### 3.3. Principal Findings

Pregnancy is the most susceptible period for getting exposed to harmful phthalates. In our study, exposure to most phthalate metabolites is associated with depression scores in pregnant women. However, high exposure to some HMWL, particularly MnBP, MCPP, MEHP, MiBP, MEOHP, and MnBP, was strongly associated with depression in pregnant women, as we hypothesized. In addition, a mixture of MCNP, MBP, MiBP, MnBP, and MEHP was also associated with depression levels in our study. These findings confirm the role of harmful phthalate exposure in depression among pregnant women. Therefore, proper prevention and monitoring strategies are required for pregnant women to avoid adverse consequences of these exposures, particularly depression leading to multiple maternal and neonatal adverse health outcomes.

### 3.4. Results in the Context of What Is Known

Several studies have identified a relationship between urine phthalate profiles and depression in different populations. One study utilized the NHANES database to analyze associations between urine phthalates and depression in the elderly population and found that MCPP, MCNP, and MBP were positively associated with depression. Another study identified a positive association between DEHP and depression in the general U.S. adult population. So far, there is only a recent study conducted by Jacobson et al. on 139 pregnant women that showed increased susceptibility to PPD in patients in relation to eight bisphenols and twenty-two phthalate exposures in the first and second trimester. However, none of these studies evaluated the association of phthalate metabolites with antenatal depression in pregnant women. Our findings add to the plethora of available literature on maternal and fetal adverse outcomes regarding phthalate exposure throughout pregnancy. For instance, a study by Shaferr et al. found a positive association between phthalate exposure and insulin resistance and gestational diabetes development in their cohort. Our research represents another endeavor to investigate the connection between exposure to phthalates and antenatal depression, a less explored maternal adverse outcome [12]. The results from our study underscore the potential importance of conducting screenings for phthalate levels, particularly HMW, in pregnant women.

Several cohort studies demonstrated the influence of phthalate metabolites on maternal and neonatal outcomes. A cross-sectional study in the Atlanta African American Maternal-Child Cohort showed an association between increased prenatal exposure to phthalates and BPA with moderate reduction in birthweight z-scores [13]. Pregnant women recruited in the Odenese Child Cohort had phthalate metabolites measured in the urine samples during their third trimester, showing an inverse association between phthalate metabolites and offspring’s neurodevelopment [14]. Maternal exposure to mono-ethyl phthalate and mono-carboxyisooctyl phthalate had been associated with low birth weight in assisted reproductive technology-conceived infants in the Jiangsu Birth Cohort Study in China. Other endocrine disruptors, such as PFAS, have not been associated with post-partum depression at six weeks postpartum [15]. Another study by Welch et al. [16] that pooled 16 US cohorts with a total of 6045 participants showed an increased risk of preterm birth among pregnant patients with phthalate exposure. In another study done on 2174 pregnant individuals, there was no association between prenatal exposure to nonpersistent environmental chemicals and postpartum depression symptoms, except for high molecular weight phthalates for sensitive and specific postpartum definitions, specifically [17]. All maternal and neonatal outcomes have been shown to be associated with antenatal and postnatal depression.

### 3.5. Pathophysiology

Although the underlying mechanisms for the association between phthalates and depression are not fully understood, evidence reflects that phthalates have the potential to affect the hypothalamic–pituitary–gonadal (HPG) axis, affecting sex hormones and subsequently yielding adverse mental health outcomes. A study evaluated the effect of these metabolites specifically on steroidogenesis in rats and found an association between exposure and decreased hormone levels [18]. The disruption of hormones such as estrogen, androgens, and thyroid hormones, crucial for various physiological processes such as pregnancy, can have significant consequences on the pregnancy and neonatal outcomes [19]. Several mouse studies confirmed the association between phthalate exposure, abnormal maternal behavior, and adverse outcomes of maternal-pup relation development [6]. These nonpersistent chemicals have the ability to penetrate biological membranes, and the high-fat content of human milk can serve as an efficient vehicle for transporting lipophilic substances. Consequently, this phenomenon raises concerns regarding the potential impact on breastfeeding infants [20].

### 3.6. Implications for Practice and/or Policy

Collectively, our findings prompt the hypothesis that prenatal exposure to these phthalates is positively associated with higher depression scores. Moreover, exposure to phthalate metabolites during pregnancy may aggravate the influence of antenatal depression. Considering the capability of phthalates to traverse the placenta and transfer to the fetus, as well as their potential presence in breast milk, it is crucial to acknowledge that maternal phthalate exposure during pregnancy can exert effects not only on the mother but also on the developing fetus and the breastfeeding newborn. It is also important to highlight that earlier research has demonstrated the limited effectiveness of the placenta as a barrier against Endocrine-Disrupting Chemicals (EDCs) [21]. Additionally, fetal organs exhibit heightened sensitivity to EDC exposure compared to adults. In one mouse study, perinatal exposure to phthalates affected depression-like behaviors in the offspring of the phthalate-exposed mice [22]. Another study showed that women’s exposure to phthalates could result in changes in the epigenetics with antiandrogenic effects, such as decreased penile size in the neonates [23]. Our study is another attempt to highlight the effect of the widely available EDCs, especially phthalates, on pregnancy and its association with antenatal depression. In contrast to the postpartum depression and phthalate exposure literature, further studies are needed to elucidate the causal relationship between exposure and antenatal depression.

## 4. Strengths and Limitations

The study is constrained by several limitations, including a small sample size despite including multiple NHANES datasets from multiple years, the absence of psychiatric evaluation or clinical diagnosis, and the cross-sectional design employed. We note that our findings should be interpreted in the context of the study design. Larger longitudinal studies are needed to both confirm these findings and identify whether other phthalates are associated with mild to severe depression in pregnant women. Our study has a single-point exposure; the timing of exposure is unknown, and we could not determine whether the subjects were exposed early or late in their pregnancy to the above phthalates. The results of this study underscore the potential for phthalate metabolites to impact hormonally mediated health outcomes in pregnant women. This study underscores the biological plausibility of phthalate-induced sex steroid hormone disruption as a pathway contributing to the pathogenesis of PPD. It also emphasizes the significance of the pregnancy period as a crucial window for exposure to EDCs.

Our study boasts various strengths. The data were sourced from the latest cycles of NHANES, wherein phthalate levels and self-reported depression during pregnancy were documented within a well-established, population-based database. We included the women who currently reported being pregnant and studied the phthalate metabolites in pregnant women, which are often understudied. We also validated our findings with a categorized distribution of phthalates and depression scores. We also evaluated the association of the composite mixture of phthalate compounds with antenatal depression. Our study is the first to associate the markers of mild to severe depression in pregnant females with exposures to phthalate metabolites. Considering a higher prevalence of depression in females, minimizing exposure to environmental stressors such as phthalates could be a potential strategy for reducing adverse health outcomes, including depression, particularly in susceptible populations like pregnant women.

## 5. Conclusions

Our findings are concordant with the recent findings of the association of phthalate metabolites with postnatal depression in pregnant women. Our study demonstrates an association between exposure to phthalates and elevated depression scores, as well as increased prevalence of depressive disorder in pregnant women. There is an increased need for community and public awareness to avoid products containing plasticizers. There should be a routine screening for phthalate exposure and antenatal depression, at least among the highly exposed pregnant women. Given the ubiquity of phthalates in the environment, future longitudinal studies are needed to validate our observations.

## Figures and Tables

**Figure 1 toxics-13-00838-f001:**
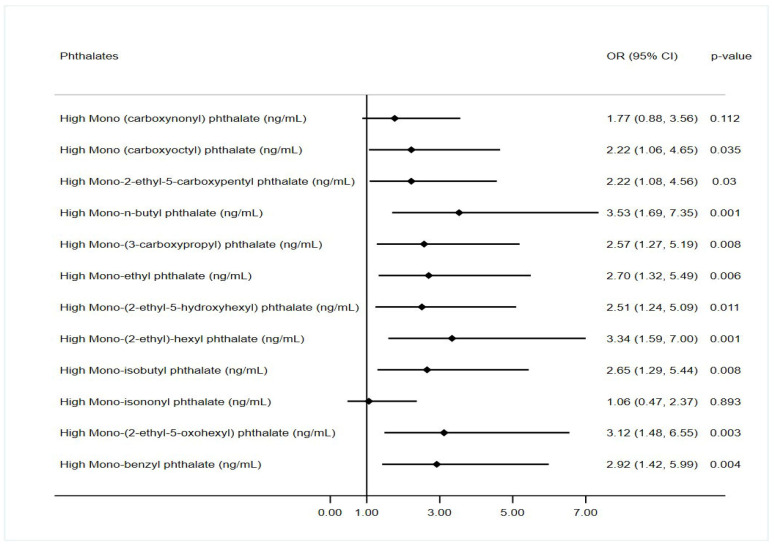
Association of phthalate concentration with depression scores.

**Table 1 toxics-13-00838-t001:** Depression scores according to baseline characteristics of pregnant women.

Factor	Value	Depression Score	*p*-Value
N	208	Median (IQR)	
Age			
18–30 years	145 (69.71%)	3.00 (1.00, 5.00)	0.37
≥31 years	63 (30.29%)	3.00 (1.00, 6.00)	
Marital Status			
Married	130 (62.50%)	3.00 (1.00, 5.00)	0.26
Others	78 (37.50%)	3.00 (1.00, 7.00)	
Ethnicity			
Hispanic	13 (6.25%)	2.00 (2.00, 4.00)	0.70
Non-Hispanic	31 (14.90%)	3.00 (1.00, 6.00)	
Others/Unknown/missing	164 (78.85%)	3.00 (1.00, 5.00)	
Income			
0 to 44,999	48 (23.08%)	4.00 (1.00, 6.50)	0.18
45,000 to 99,999	26 (12.50%)	2.00 (1.00, 4.00)	
100,000 and above	13 (6.25%)	2.00 (1.00, 3.00)	
Unknown	121 (58.17%)	3.00 (1.00, 5.00)	
Education			
less than 9th grade	18 (8.65%)	2.00 (0.00, 5.00)	0.077
9th to 12th grade	90 (43.27%)	4.00 (1.00, 7.00)	
college or above	100 (48.08%)	2.00 (1.00, 4.00)	
US born			
US born	37 (17.79%)	3.00 (1.00, 5.00)	0.17
Non US born	19 (9.13%)	2.00 (1.00, 3.00)	
Unknown	152 (73.08%)	3.00 (1.00, 5.50)	
Ever alcohol			
No	49 (23.56%)	2.00 (1.00, 5.00)	0.17
Yes	137 (65.87%)	3.00 (1.00, 6.00)	
Unknown	22 (10.58%)	3.50 (2.00, 6.00)	
Smoking Status			
No	145 (69.71%)	3.00 (1.00, 5.00)	0.071
Yes	53 (25.48%)	4.00 (1.00, 7.00)	
Unknown	10 (4.81%)	3.00 (2.00, 8.00)	
Physical activity			
No	64 (30.77%)	3.00 (1.00, 5.00)	0.41
Yes	32 (15.38%)	3.50 (1.50, 6.00)	
Unknown	112 (53.85%)	3.00 (1.00, 5.00)	
Obesity			
No	134 (64.73%)	3.00 (1.00, 5.00)	0.44
Yes	73 (35.27%)	3.00 (1.00, 6.00)	

Abbreviations: US, The United State; IQR, Interquartile Range.

**Table 2 toxics-13-00838-t002:** Unadjusted association of each phthalate concentration with depression scores.

Phthalates	Median (IQR)	RC * (95% CI)	*p*-Value
Mono (carboxynonyl) phthalate (ng/mL)	1.88 (1.04, 3.76)	0.13 (0.03, 0.24)	0.014
Mono (carboxyoctyl) phthalate (ng/mL)	4.75 (2.60, 11.70)	0.06 (−0.02, 0.14)	0.158
Mono-2-ethyl-5-carboxypentyl phthalate (ng/mL)	15.45 (6.60, 34.60)	0.08 (0.00, 0.17)	0.042
Mono-n-butyl phthalate (ng/mL)	14.70 (7.05, 31.90)	0.17 (0.09, 0.26)	<0.001
Mono-(3-carboxypropyl) phthalate (ng/mL)	1.60 (0.70, 2.87)	0.16 (0.07, 0.26)	0.001
Mono-ethyl phthalate (ng/mL)	59.95 (21.41, 228.69)	0.07 (0.00, 0.13)	0.045
Mono-(2-ethyl-5-hydroxyhexyl) phthalate (ng/mL)	9.95 (3.85, 21.35)	0.08 (0.01, 0.16)	0.029
Mono-(2-ethyl)-hexyl phthalate (ng/mL)	1.92 (0.85, 5.20)	0.08 (0.00, 0.16)	0.049
Mono-isobutyl phthalate (ng/mL)	6.40 (2.50, 11.95)	0.12 (0.03, 0.21)	0.006
Mono-isononyl phthalate (ng/mL)	0.87 (0.67, 0.87)	0.03 (−0.10, 0.16)	0.644
Mono-(2-ethyl-5-oxohexyl) phthalate (ng/mL)	8.20 (3.55, 15.90)	0.09 (0.02, 0.17)	0.018
Mono-benzyl phthalate (ng/mL)	6.48 (2.68, 16.37)	0.14 (0.06, 0.22)	<0.001

Abbreviations: IQR, Interquartile Range; RC, Regression Coefficient; CI: Confidence Interval. * Log-transformed values of depression scores and phthalate concentrations were included in the analyses.

**Table 3 toxics-13-00838-t003:** Adjusted association of each phthalate metabolite concentration with depression scores.

Phthalates	aRC * (95% CI)	*p*-Value
Mono (carboxynonyl) phthalate (ng/mL)	0.11 (0.00, 0.22)	0.055
Mono (carboxyoctyl) phthalate (ng/mL)	0.05 (−0.03, 0.14)	0.226
Mono-2-ethyl-5-carboxypentyl phthalate (ng/mL)	0.10 (0.01, 0.18)	0.030
Mono-n-butyl phthalate (ng/mL)	0.17 (0.07, 0.26)	0.001
Mono-(3-carboxypropyl) phthalate (ng/mL)	0.15 (0.05, 0.25)	0.003
Mono-ethyl phthalate (ng/mL)	0.07 (0.00, 0.14)	0.046
Mono-(2-ethyl-5-hydroxyhexyl) phthalate (ng/mL)	0.09 (0.01, 0.16)	0.033
Mono-(2-ethyl)-hexyl phthalate (ng/mL)	0.08 (0.00, 0.17)	0.049
Mono-isobutyl phthalate (ng/mL)	0.10 (0.01, 0.19)	0.037
Mono-isononyl phthalate (ng/mL)	0.05 (−0.09, 0.19)	0.472
Mono-(2-ethyl-5-oxohexyl) phthalate (ng/mL)	0.10 (0.02, 0.18)	0.018
Mono-benzyl phthalate (ng/mL)	0.13 (0.04, 0.21)	0.004

Abbreviations: aRC, Adjusted Regression Coefficient; CI: Confidence Interval. * Log-transformed values of depression scores and phthalate concentrations were included in the analyses. Analyses were adjusted for age, ethnicity/race, education, marital status, income status, smoking, alcohol use, physical activity, and obesity.

**Table 4 toxics-13-00838-t004:** Unadjusted and adjusted association of composite phthalate metabolite concentration with depression scores.

Factor	Exposure	RC (95% CI)	*p*-Value
**Primary analysis**			
Unadjusted analysis	Composite scores of all phthalate metabolites	0.30 (0.13, 0.47)	<0.001
Adjusted analysis	Composite scores of all phthalate metabolites	0.22 (0.03, 0.41)	0.029
**Validation analysis**			
Unadjusted analysis	Composite scores of selected phthalates metabolites	0.29 (0.13, 0.45)	<0.001
Adjusted analysis	Composite scores of selected phthalate metabolites	0.18 (0.00, 0.36)	0.049
Adjusted analysis	Composite scores of selected phthalate metabolites	0.25 (0.12, 0.38)	<0.001

Abbreviations: WQS: Weighted Quantile Sum Regression; QGCOMP: Quantile G-Computation Model; CI: Confidence Interval. Selected phthalates include MCNP, MBP, MiBP, MnBP, and MEHP obtained from unadjusted WQS regression. Log-transformed values of depression scores and phthalate concentrations were included in the analyses. Analyses were adjusted for age, ethnicity/race, education, marital status, income status, smoking, alcohol use, physical activity, and obesity.

## Data Availability

No new data were created or analyzed in this study. Data sharing is not applicable to this article.

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
