# Peer review of "Association of Exposure to Phthalate Metabolites with Antenatal Depression in US Pregnant Women"

_toxics, 2025, doi:10.3390/toxics13100838_

Round 1

Reviewer 1 Report

Comments and Suggestions for Authors

The authors investigated the association between phthalate metabolites and antenatal depression among 208 pregnant women from the NHANES 2005–2018 survey. However, something should be clarified before publication. My comments are as follows:

1) Although NHANES 2005–2018 included 17,190 women, only 208 pregnant women were included in this study. How well do these 208 women represent the broader U.S. population? Given potential differences in demographic characteristics, how did the authors effectively control for non-pollutant factors that may influence depression?

2) The manuscript suggests that phthalates may affect mental health outcomes by disrupting the hypothalamic-pituitary-gonadal (HPG) axis and influencing sex hormones. However, the subsequent discussion of results does not sufficiently support this claim. Please provide stronger evidence to reinforce this proposed mechanism.

Author Response

The authors investigated the association between phthalate metabolites and antenatal depression among 208 pregnant women from the NHANES 2005–2018 survey. However, something should be clarified before publication. My comments are as follows:

1) Although NHANES 2005–2018 included 17,190 women, only 208 pregnant women were included in this study. How well do these 208 women represent the broader U.S. population? Given potential differences in demographic characteristics, how did the authors effectively control for non-pollutant factors that may influence depression?

Response: We sincerely thank the reviewer for their comments and suggestions. We agree that the limited sample size is an important and relevant consideration. From 17, 190 women, we only included the women who met the following criteria: 

a).  Women who reported they were currently pregnant at the time of the sample collection. 

b). Women who had the phthalates data collected. 

Although the sample size for women was fairly large, 17,190 women and 604 pregnant women, our sample size shrank to 208 as only one third of the pregnant women had all the phthalates we were looking for. To maintain the consistency of the data, we limited the sample to 208. 

2) The manuscript suggests that phthalates may affect mental health outcomes by disrupting the hypothalamic-pituitary-gonadal (HPG) axis and influencing sex hormones. However, the subsequent discussion of results does not sufficiently support this claim. Please provide stronger evidence to reinforce this proposed mechanism.

Response: We thank the reviewer for this important observation. Following your suggestions, we have updated the pathophysiology section and updated the citations to include experimental evidence showing that phthalates impair steroidogenesis in animal models, leading to decreased ovarian hormone production. We also now highlight rodent studies demonstrating that perinatal phthalate exposure leads to abnormal maternal behavior and depression-like phenotypes in offspring. We revised the language throughout the manuscript to emphasize that we only suggest association and not causality. 

Reviewer 2 Report

Comments and Suggestions for Authors

This is a theme of high interest as both endocrine disruption and pregnancy outcomes and hot topics in endocrinology, but other specialties aswell. Data is clearly presented, but the hypothesis should be brought earlier in the introduction .

Results are clearly shown, but some tables which are large and make the paper look messy, could be moved to the Supplementary.

You could clarify methodological choices and temper the "causality" statements, downgrading its to association or potential interference. 

It would be nice to widen and elaborate the call for action from the public health perspective.

Author Response

1. This is a theme of high interest as both endocrine disruption and pregnancy outcomes and hot topics in endocrinology, but other specialties aswell. Data is clearly presented, but the hypothesis should be brought earlier in the introduction . 

Response: We want to sincerely thank the reviewer for their attention and meticulous review. We have moved the hyposis towards the beginning of the manuscript based on your suggestion. 

2. Results are clearly shown, but some tables, which are large and make the paper look messy, could be moved to the Supplementary.

Response: We agree with the suggestions. To improve readability, we have moved some tables to the supplementary section and also updated the legends. 

3. You could clarify methodological choices and temper the "causality" statements, downgrading its to association or potential interference. 

Response: We agree with this suggestion and have now revised the manuscript thoroughly to suggest association and not causality. Thank you for pointing it out. We have also updated the methods section, indicating that we have done a cross-sectional study and secondary data analysis for NHANES data. 

4. It would be beneficial to expand and elaborate on the call for action from a public health perspective.

Response: We agree and have updated the policy statement in section 3.6. We have emphasized the universalness of phthalates, their burden, levels of expression, and the importance of community awareness, product regulation, and the need for routine monitoring. 

Reviewer 3 Report

Comments and Suggestions for Authors

Okay, author's kudos on an interestng and novel cross-sectional study design.  I like your paper, enjoyed reading it and want to see you make it even more useful for future researchers...perhaps even for attorneys who might read it in a toxic tort case.  So I have a number of suggestions that will improve it.

  1. The writing was wonderful overall.  Here are a few minor but important details.  I think the journal prefers U.S. instead of US (title and other places).
  2. Space at line 13-14, from2005...needs a space
  3. Line 110-111...The sum of 9 item..."The sum of the 9 items..."
  4. Line 300-301...Despite including of multiple NHANES datasets..."Despite inclusion of multiple NHANES datasets...
  5. Line 309...you have a hyphen that should not be in biological (no hyphen needed)

Methods -  You need to very clearly own that this is a cross-sectional study...it is a little unclear that you did not run the phthalate analyses, make that very clear. At least my impression is you took data already available from NHANES. If your team was involved in analyzing that data directly make it more apparent.

Also make it very clear how long data acquisition continued.  You note the two time periods but my concern is that this is a period of 9 years..so while you have paired pregnancies and single urine samples (make sure to not that is a weakness also) they occur over periods of time that could impact how the populations is impacted by world events, other exposures. Was there any way that you guarded against between-cycle data time confounding, perhaps statistically can you check that to make sure that earlier and later data are not dramatically different and discuss that a bit?  Just be very clear and upfront about that.

Discussion - remember to make sure that your language reflects association and not certain causality.  Watch the claims that are made, this is cross-sectional.  It is very interesting and suggests additional work is needed, but it is not definitive.

Graphs and Figures: Revisit all of your graph and figure titles and legends.  Some are rather vague...understandable exhaustion, as we all usually name them after slogging through the real "body" of the paper.  But now that you have some time, make them as descriptive as the space considerations of the journal will allow.  This is for Title II compliance for disabled and poorly sighted readers.  The only real issue I have is with Supplemental tables lines 406-420...a bit hard to distinguish between groups...perhaps pre-publishing checks can take care of this and some formatting can improve it?

Author Response

Okay, author's kudos on an interesting and novel cross-sectional study design.  I like your paper, enjoyed reading it, and want to see you make it even more useful for future researchers...perhaps even for attorneys who might read it in a toxic tort case.  So I have a number of suggestions that will improve it.

Response to reviewers: Many thanks for your kind comments. We want to emphasize the importance of both toxicant screening and depression screening during pregnancy to improve pregnancy outcomes for mother and child, and also subsequent pregnancies. Your kind comments have encouraged us more in our efforts. 

  1. The writing was wonderful overall.  Here are a few minor but important details.  I think the journal prefers U.S. instead of US (title and other places). Response: Corrected as per the suggestion 
  2. Space at line 13-14, from2005...needs a space. Response: Corrected as per the suggestion 
  3. Line 110-111...The sum of 9 item..."The sum of the 9 items..." Response: Corrected as per the suggestion 
  4. Line 300-301...Despite including of multiple NHANES datasets..."Despite the inclusion of multiple NHANES datasets... Response: Corrected as per the suggestion 
  5. Line 309...you have a hyphen that should not be in biological (no hyphen needed). Response: Corrected as per the suggestion 

Methods -  You need to very clearly own that this is a cross-sectional study...it is a little unclear that you did not run the phthalate analyses, make that very clear. At least my impression is you took data already available from NHANES. If your team was involved in analyzing that data directly make it more apparent.

Response: We appreciate your kind comment. We have updated the nature of the study as cross-sectional and also updated that this is a secondary analysis of the NHANES database. 

Also make it very clear how long data acquisition continued.  You note the two time periods but my concern is that this is a period of 9 years..so while you have paired pregnancies and single urine samples (make sure to not that is a weakness also) they occur over periods of time that could impact how the populations is impacted by world events, other exposures. Was there any way that you guarded against between-cycle data time confounding, perhaps statistically can you check that to make sure that earlier and later data are not dramatically different and discuss that a bit?  Just be very clear and upfront about that.

Response: Thank you for your great comment. This is a genuine concern with pooled data collected over time, especially in cross-sectional studies. To overcome this, for all NAHNES data, we adjusted the year of cycle as a categorical variable; however, despite this adjustment, we did not see any change in our results. We also ideally use NHANES survey weights for pooled analyses (with proper weight adjustment across cycles).

Discussion - remember to make sure that your language reflects association and not certain causality.  Watch the claims that are made, this is cross-sectional.  It is very interesting and suggests additional work is needed, but it is not definitive.

Response: Throughout the manuscript, we have made this change. 

Graphs and Figures: Revisit all of your graph and figure titles and legends.  Some are rather vague...understandable exhaustion, as we all usually name them after slogging through the real "body" of the paper.  But now that you have some time, make them as descriptive as the space considerations of the journal will allow.  This is for Title II compliance for disabled and poorly sighted readers.  The only real issue I have is with Supplemental tables lines 406-420...a bit hard to distinguish between groups...perhaps pre-publishing checks can take care of this and some formatting can improve it?

Thank you: We have moved some tables to the supplementary section and also simplified the legends for improved readibility